# Intravenous Corticosteroid Therapy for Acute Lumbar Radicular Pain

**DOI:** 10.3390/jcm11175127

**Published:** 2022-08-31

**Authors:** Oded Hershkovich, Yaakov Mor, Raphael Lotan

**Affiliations:** 1Department of Orthopedic Surgery, Wolfson Medical Center, Ha-Lokhamim St. 62, Holon 5822012, Israel; 2Sackler Faculty of Medicine, Tel Aviv University, Tel Aviv 6997801, Israel

**Keywords:** sciatica, radiculopathy, herniated disc, steroid treatment, pharmacotherapy

## Abstract

Introduction: The efficacy of pharmacological interventions for acute lumbar radicular pain (ALRP) is limited, and systemic steroid use remains controversial. We evaluated the effectiveness and tolerance of systemic steroid use in a cohort of patients with ALRP. Methods: A retrospective cohort of 56 patients (including 24 females) were admitted with intractable ALRP resistance to conservative treatment of NSAIDs and opiates between the years 2016 and 2018. Medical records were studied for demographics, physical examination findings, Visual Analogue Score (VAS), IV steroids side effects, and recent imaging findings. All patients received a daily dose of IV 24 mg Dexamethasone until discharge, SNRB, or surgery. Results: The average IV steroid treatment was 3.9 (±2.8) days, with most patients showing significant pain relief allowing discharge (69.7%). SNRB was required in 19.6% and surgical intervention in 10.7% within the same admission. Multivariate analysis did not find any parameter to predict treatment failure (age, gender, motor/sensory deficit, CT/MRI findings). The motor deficit, positive straight leg raising (SLR), and dural sac compression on CT were higher in the intervention group but did not reach statistical significance. One patient required discontinuation of IV steroids due to elevated blood pressure. Conclusions: Despite the insufficient evidence in the literature, IV steroid treatment is still a viable option in ALRP treatment, with pain relief allowing discharge in 70% of patients and a low complication rate. Our study found daily 24 mg IV dexamethasone for ALRP to be an effective treatment and helpful in most patients admitted. This study supports the common practice used by spine units.

## 1. Introduction

Acute lumbar radicular pain (ALRP), also known as sciatica, is a widespread clinical complaint with an annual incidence of about 0.5% in Western adult populations [1]. ALRP is characterized by radiating buttock and leg pain in a lumbar nerve root distribution [2,3]. It is frequently associated with herniated nucleus pulposus (HNP) [4,5], with a lifetime incidence beyond 10% [6] and a considerable bearing on health resources [7]. Spine pathologies are a leading source of disability and work absenteeism and have a massive financial burden [8]. The pathophysiology of ALRP due to HNP relates to nerve root compression and a proinflammatory cascade with the release of molecules such as IL-1b, IL-6, TNF-α, nitric oxide, and prostaglandin E2 [9]. Recent studies showed the importance of Th17 lymphocytes in the inflammatory cascade [10]. Spontaneous recovery occurs in most patients; however, many bear considerable pain and disability during recovery [3,4,11]. Non-steroidal anti-inflammatory drugs (NSAIDs) are usually the first-line drug of choice; steroids are considered an alternative that can be used systematically or locally [12]. Among patients with ALRP due to a herniated lumbar disk, a short course of oral steroids, compared with a placebo, resulted in modestly improved function but no improvement in pain [13]. If symptoms persist, invasive procedures such as selective nerve root block (SNRB) with steroids and local anesthetics are widely used, frequently followed by lumbar discectomy [4,14,15,16]. Treatment that accelerates recovery will substantially benefit patients and reduce the need for expensive, invasive procedures. Despite inconsistent evidence, SNRBs are frequently offered under the hypothesis that ALRP symptoms are caused by the inflammation of the affected lumbar nerve root [5,17,18,19]. The use of epidural administration in ALRP is still controversial [4,20]. Green et al. [21] reported encouraging results on pain relief by systemic steroids using an intramuscular tapering dose of Dexamethasone. Previous systematic literature reviews have suggested a limited role of systemic steroids in ALRP [12,22]. The literature regarding the efficacy of epidural, intramuscular, or oral glucocorticoids in acute sciatica has provided conflicting results [4]. Some have shown a short-term advantage [23,24], while others have not shown any value [4]. The rationale for intravenous (IV) glucocorticoid administration in acute sciatica is rapid delivery of high-level glucocorticoids that reduce the inflammatory cascade caused by the HNP without the discomfort and dangers of epidural injection and allow an easy and accessible treatment as SNRB availability varies between countries and health systems [25,26,27]. Previous studies suggested that IV administration has a better risk–benefit ratio than other methods of treatment [28,29]; still, recent publications showed limited effect [12,30]. Despite numerous publications regarding IV glucocorticoid administration in ALRP, the role and efficacy of this treatment remain undetermined without an agreed single treatment protocol or dosage showing superiority [25].

Our study aimed to examine the outcome of our local long-used protocol of IV glucocorticoid administration in ALRP regarding admission time, pain relief effect, and the need for further invasive procedures (SNRB and Micro-discectomy).

## 2. Methods

We conducted a retrospective cohort study of all ALRP patients admitted to the orthopedic surgery department between the years 2016 to 2018. Records of patients admitted directly from the emergency department (ER) due to acute LBP with radiculopathy were examined.

Inclusion criteria included patients suffering from acute ongoing radicular pain lasting from one day to six months in the ER that failed treatment with an intramuscular Meperidine, an intramuscular Diclofenac, and an intravenous Assival with a recent CT or MRI (less than six months from admission). Those patients with non-improving ALRP were admitted for steroid treatment in our department.

Patients not suitable for steroidal treatment were excluded due to uncontrolled diabetes, hypertension, anticoagulant therapy, systemic infection, malignancy, and patients that refused steroidal treatment. Elective admissions for discectomies or selective nerve root injections were excluded as well. Patients diagnosed with cauda equina syndrome or new motor deficits (less than 48 h) were operated on urgently and thus excluded.

The study cohort’s patient records were reviewed for admission time and pre-admission pain treatment (NSAIDs and Opiates) versus treatment needed after IV steroids administration. During the hospitalization period, visual Analogue Score (VAS), and physical examination findings, including motor and sensor deficit by root, reflex grading, and straight leg raising test (SLR), were also collected and summed.

Two fellowship-trained spine surgeons reviewed the cohort’s imaging studies (CT and/or MRI available) and classified findings by etiology (herniated disc versus degenerative stenosis), maximal area of nerve compression (central canal, recess, foraminal, and extraforaminal), and the degree of pressure (contact, light pressure, severe pressure). All parameters were scored by three consultants (two experienced spinal surgeons and one musculoskeletal radiologist).

Our long-used orthopedic department’s treatment protocol for intractable ALRP leading to patient admission includes a single daily dose of intravenous 24 mg of Dexamethasone for 2–7 days, depending on the patient’s response to treatment. This Dexamethasone dosage is an empirical historical dosage. Patients that improve under this treatment protocol are discharged with further treatment with NSAIDs and opiates as needed. Patients that show gradual improvement continue treatment as needed for a maximum of 7 days. Patients resistant to treatment are offered an SNRB at the pain clinic within admission unless they develop motor deficits. Patients that do not improve under SNRB or suffer from progressive motor deficits underwent surgery. Iv Dexamethasone was continued until the patient received SNRB or surgery and up to 14 days.

Dexamethasone dosage is constant and unrelated to patient weight since a 24 mg daily dosage falls within the therapeutic dosage of less than 0.5 mg/kg, even in low-BMI patients.

Patient response to IV steroids treatment was recorded. Treatment success was defined as sufficient pain relief to allow patient discharge, and failure as a lack of pain relief that required SNRB or surgical intervention during the same admission. IV steroid side effects were collected (uncontrolled hypertension, diabetic ketoacidosis, gastrointestinal bleeding, cerebrovascular accident, or cardiac event).

Statistical analysis was performed using the SAS Software, Version 9.4. Mean ± Std or median and IQR were used to present continuous variables; categorical variables were presented by (N, %). An event was defined as surgery or SNRB before release. The Cox Proportional Hazards model was used to assess pathological anatomy on imaging and treatment results.

Our Medical Center’s IRB committee approved the study.

## 3. Results

The study cohort included 56 patients (24 females, 32 males); the average age was 57.8 years (21–89 years). The descriptive data are presented in Table 1. Intravenous Dexamethasone (IV) treatment time was 4.05 days (±2.61) with a range of 2–14 days. The IV course was slightly shorter in males (3.72 days) compared to females (4.21 days; *p* = 0.271). Out of the 56 patients receiving Dexamethasone, 17 (10 females and 7 males) failed the conservative treatment, 11 patients (19.6%) underwent SNRB in admission, and 6 patients (10.7%) underwent surgery; none of these patients deteriorated neurologically (Table 1).

Most of our cohort (69.6%) had significant pain relief under the Dexamethasone treatment protocol, which allowed discharge without further intervention.

Further investigating our cohort’s clinical and radiographic parameters, we could not identify any parameter predicting the success or failure of IV Dexamethasone treatment for ALRP (Table 2). Parameters examined included motor and sensory deficits, SLR response, location, cause and extent of root compression on imaging studies (Table 1). Although some parameters were associated with failure of treatment and the need for further invasive procedures (such as positive SLR; OR = 2.276) and severe stenosis (OR = 2.295), none reached statistical significance, probably due to our small cohort of patients (Table 2).

Although Dexamethasone is a powerful glucocorticoid, even a minimal but continuous dosage of 0.5 mg/kg daily can lead to stress on the adrenal glands; a routine administration of antacids and anticholinergic drugs was not used routinely to prevent gastrointestinal ulcers, hypertension, and bleeding diathesis. We did not encounter any adverse events in our cohort other than one patient that developed elevated blood pressure and required steroid treatment discontinuation. However, this patient improved enough to warrant discharge after five days of admission without further interventions.

All patients were followed up in the outpatient clinic. We did not have re-admissions within three months of patients successfully treated with the IV steroids protocol.

## 4. Discussion

Our study shows the benefit of once-daily 24 mg of intravenous Dexamethasone in treating early-phase ALRP. Although relatively small, in our cohort, 70% of the patients improved on average within 3.9 days of admission to a level of pain that allowed discharge without needing an SNRB or surgery.

ALRP associated with a herniated nucleus pulposus frequently triggers considerable pain and disability, with a significant economic burden [4,5,8]. Therapy alternatives include medical advice and education, self-care, oral medications) such as NSAIDs) and oral steroids, followed by various physical modalities (such as physiotherapy, ultrasound, electrical stimulation, etc.). Interventional procedures, including epidural steroid injections, SNRBs, and microdiscectomy surgery, are warranted for intractable radicular pain [4,14,15,22,31].

Over the years, multiple studies have examined systemic steroids’ role in treating patients with ALRP [30,32,33,34,35,36]. These studies have examined small cohorts with low statistical power [12,33,34,36] and did not find steroid treatment efficacious. On the other hand, this treatment significantly increased side effects, including an increased risk for surgery [37,38]. However, current research (with a larger cohort) found a trend toward significant pain relief in early ALRP following a single IM administration of Methylprednisolone [35]. Goldberg et al. also reported a slight, statistically significant improvement in function (as measured by the ODI) at 3 and 52 weeks following Prednisone treatment protocol, but with no difference in lower extremity pain scores [13]. In many medical centers, IV steroids are used based on local protocol and experience.

This study included ALRP patients resistant to major initial treatment in the ER (opiates, NSAIDS, and muscle relaxants). ALRP usually resolves within weeks to months [39,40]; studies have shown ALRP improvement under rest, physiotherapy, and pain control over several weeks of treatment. Adding IV steroids or SNRB can shorten the severe acute pain period [12,13,15]. As hospitalization beds are a valuable health resource, the treatment goal is to gain fast pain control to enable discharge. This is why an immediate IV steroid protocol was initiated to treat ALRP patients in our unit.

Once-daily, 24 mg of intravenous Dexamethasone protocol successfully treated 70% of ALRP patients hospitalized due to intractable pain in our cohort. Patients improved on average within 3.9 days of treatment to a level of pain that allowed discharge without needing an SNRB or surgery. Seventeen patients (30.3%) required further treatment, eleven improved with SNRB, and six required micro-discectomies. There were no readmissions up to 3 months following successful IV steroid treatment. We had one uncontrolled elevated blood pressure case that mandated discontinuation of therapy but with pain relief that permitted discharge.

Even today, medical resources are limited worldwide, and portions of the population do not enjoy accessibility to specialized pain clinics. Other patients cannot undergo spinal injections due to anticoagulation treatment but require urgent pain relief. A simple and available treatment protocol that provides an early treatment option for ALRP with similar results to SNRBs [35,36] is required. The once-daily treatment protocol of intravenous 24 mg Dexamethasone is simple (one dose for all), available, cheap, and without significant adverse events, achieving 70% of early pain relief. This protocol is ideal for general practitioners, rural areas, and hospitals lacking pain services.

Our study’s main limitations include a small nonrandomized cohort and a retrospective nature. The relatively small cohort might explain the low complication rate reported. We did not compare oral steroids treatment vs. our IV protocol, which could simplify treatment further.

In conclusion, this study supports the practice of IV steroids use in treating ALRP, although the use of IV steroids remains relatively unexplored. This study found that a short-term IV steroid treatment protocol for ARLP achieved 70% successful hospitalization discharge and low complication rates. Further randomized controlled studies of intractable ALRP are required to substantiate the preferred protocol and its adverse effects.

## Figures and Tables

**Table 1 jcm-11-05127-t001:** Cohort Demographics (N = 56).

*Variables*	Conservative (*n* = 39)	SNRB (*n* = 11)	Surgery (*n* = 6)	*p*-Value
** *Age, Average (Yr.)* **	55.6 ± 18.4	66.1 ± 14.6	57.3 ± 13.7	0.103
** *Gender Female (%)* **	35.9	63.6	50.0	0.149
** *IV Dexamethasone Treatment Days* **	3.9 ± 2.8	4.4 ± 2.1	4.5 ± 2.3	0.503
**Motor deficit (%)**	37.8	40	57.1	0.522
**Sensory deficit (%)**	16.2	30	42.9	0.117
**Positive SLR (%)**	48.6	70	42.9	0. 914
**Central canal stenosis (%)**	73	70	71.4	0.856
**Recess stenosis (%)**	10.8	10	0	0.562
**Foraminal stenosis (%)**	54.1	50.0	42.9	0.633
**Extra-foraminal stenosis (%)**	8.1	0	0	0.227
**Mild stenosis (%)**	18.9	0	0	0.055
**Moderate stenosis (%)**	56.8	70	28.6	0.793
**Severe stenosis (%)**	24.3	30	57.1	0.208

ALRP = Acute lumbar radicular pain, SNRB = Selective nerve root block, SLR = Straight leg raising.

**Table 2 jcm-11-05127-t002:** Hazard Ratios (HR).

Character	Hazard Ratio	95% Lower Confidence Limit for Hazard Ratio	95% Upper Confidence Limit for Hazard Ratio	Pr > ChiSq
Motor deficit	1.945	0.729	5.193	0.1842
Sensory deficit	1.581	0.581	4.297	0.3697
Positive SLR	2.276	0.824	6.282	0.1124
Central canal stenosis	0.783	0.295	2.075	0.6227
Recess stenosis	1.817	0.436	7.565	0.4120
Foraminal stenosis	1.505	0.553	4.097	0.4237
Extra-foraminal stenosis	0.395	0.021	7.271	0.5317
Mild stenosis	0.152	0.008	3.075	0.2198
Moderate stenosis	1.079	0.382	3.048	0.8857
Severe stenosis	2.295	0.808	6.523	0.1189
SNRB	1.371	0.337	5.570	0.6593
Steroids side effects	0.253	0.009	6.884	0.4152

ALRP = Acute lumbar radicular pain, SNRB = Selective nerve root block, SLR = Straight leg raising.

## Data Availability

The complete data is available under a confidentiality restriction.

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
