# Peer review of "Intravenous Corticosteroid Therapy for Acute Lumbar Radicular Pain"

_jcm, 2022, doi:10.3390/jcm11175127_

Round 1
Reviewer 1 Report (New Reviewer)
1. Inclusion criteria: The caption of the paper is ACUTE lumbar radicular pain. What is the definition of "ACUTE"? Does it include patients who attend the ER within hours after the onset of pain? There is a mention that CT/MRI must have been done within 6 months, which makes the definition of acute even more confusing. If the patient has pain for several months before going to the ER (or may have gone to ER more than once), calling it acute lumbar radiculopathy may not be appropriate. In a similar paper by Kovarsky (reference 27) this controversy has been avoided by using the term lumbosacral radiculopathy without qualifying it as acute. The authors could provide the duration of pain (from the onset to the date of entry into the study) in the table and see if t contributed to the outcome after steroid therapy.
2. The choice of dose of dexamethasone appears to be purely empirical. How did the surgeons determine that 20 mg IV is the ideal dose without trying other doses and establishing it as the protocol for your clinic?
Author Response
August 27, 2022
REVISION COVER LETTER
I am enclosing the revised manuscript entitled "Intravenous corticosteroid therapy for acute lumbar radicular pain" submitted to the "Journal of Clinical Medicine" after extensive revision based on your reviewers' remarks for possible evaluation.
With this manuscript's submission, I would like to undertake that the manuscript mentioned above has not been published elsewhere, accepted for publication elsewhere or under editorial review for publication elsewhere. My Institute's representative is fully aware of this submission.
Type of Submitted manuscript:
- Original Article
I want to share the following information with Editor-in-Chief:
"The efficacy of pharmacological interventions for acute lumbar radicular pain (ALRP) is limited, and systemic steroid use remains controversial. We evaluated the effectiveness and tolerance of systemic steroid use in a cohort of patients with ALRP.
In our retrospective cohort of 56 patients (24 females) admitted with intractable ALRP resistance to conservative treatment, all patients received a daily dose of IV 24 mg Dexamethasone until discharge, SNRB or Surgery.
The average IV Steroid treatment was 3.9 (±2.8) days, with most showing significant improvement enough to merit discharge (69.7%). SNRB was required in 19.6% and surgical intervention in 10.7% within the same admission. Multivariate analysis did not find any parameter to predict treatment failureand one patient required discontinuation of IV steroids due to elevated blood pressure.
Despite the insufficient evidence in the literature, IV steroid treatment is still a viable option in ALRP treatment with good pain relief (70%) and a low complication rate. Even today, medical resources are limited worldwide, and portions of the population do not enjoy accessibility to specialized pain clinics. A simple and available treatment protocol that provides an early treatment option for ALRP with similar results to SNRBs is still required. The once-daily treatment protocol of intravenous 24mg Dexamethasone is simple (one dose for all), available, cheap, and without significant adverse events achieving 70% of early pain relief. This protocol is ideal for general practitioners, rural areas, and hospitals lacking pain services and supports the common practice used in many spine units."
We believe this work's significant findings can contribute to the ongoing debate on the subject and add further knowledge and experience to the world community. As such, we would like them to be published in the journal.
Attached to this letter is our point-to-point reply below to the reviewers' comments with changes marked in our revised manuscript submitted.
On behalf of all the authors,
Sincerely yours
The corresponding author
Dear Editor
Please see our point-to-point reply below to the reviewers' comments with changes marked in our revised manuscript submitted.
We would like to appreciate the reviewer's remarks and efforts in helping us make this work better.
Review reports:
Reviewer 1 & Replies to review:
- Inclusion criteria: The caption of the paper is ACUTE lumbar radicular pain. What is the definition of "ACUTE"? Does it include patients who attend the ER within hours after the onset of pain? There is a mention that CT/MRI must have been done within 6 months, which makes the definition of acute even more confusing. If the patient has pain for several months before going to the ER (or may have gone to ER more than once), calling it acute lumbar radiculopathy may not be appropriate. In a similar paper by Kovarsky (reference 27) this controversy has been avoided by using the term lumbosacral radiculopathy without qualifying it as acute. The authors could provide the duration of pain (from the onset to the date of entry into the study) in the table and see if t contributed to the outcome after steroid therapy.
We included all patients with acute ongoing lumbar radicular pain lasting from the same day to six months, as described in the methods.
We have made this point clearer now in the text.
Change is bolded and underlined.
Failure of treatment in the ER was considered as severe acute lumbar radicular pain. Unfortunately, accurate data regarding pre-hospital symptom duration is lacking. Therefore analysis of its impact on steroid treatment results is unavailable.
- The choice of dose of Dexamethasone appears to be purely empirical. How did the surgeons determine that 20 mg IV is the ideal dose without trying other doses and establishing it as the protocol for your clinic?
We agree 24 mg IV dose of Dexamethasone is an empirical historical dosage used in our department for many years. We studied the results of this protocol.
Reviewer 2 & Replies to review:
The paper is interesting, but it comes with a hasty draft.
Since it can be of considerable interest to those who deal with ALRP even in non-specialist centers, it is important to clarify some passages better.
-page 5 In the Method, you talk about CT and MRI as recent exams, without specifying anything else. It would be useful to explain (if so) that it was the tests that led to the formulation of Table 1, which lists the status of relationships of the lumbar spine and root structures.
Please see the methods paragraph as attached here:
Two fellowship-trained spine surgeons reviewed the cohort's imaging studies (CT and/or MRI available) and classified findings by etiology (Herniated disc versus degenerative stenosis), maximal area of nerve compression (central canal, recess, foraminal and extraforaminal) and the degree of pressure (contact, light pressure, severe pressure). All parameters were scored by three consultants (two experienced spinal surgeons and one musculoskeletal radiologist).
Also:
In the results section, we added a referral to the results gathered from those studies in Table 1.
-page 6 The administration and dosages of Dexamethasone are always the same and continuous up to a maximum of 14 days. Have you ever had an administration with progressive decrease?
The protocol is 24mg single dose daily for 2-7 days without a dose change. No patient had a progressive decrease in dosage.
As for the dosages, it is certainly known that Dexamethasone, in emergency conditions (example: cerebral edema) can be administered in very high dosages.
However, since it is a powerful glucocorticoid, even a minimal but continuous dosage of 0.5 mg / kg per day can lead to stress on the adrenal glands.
It would be important to report if antacid and anticholinergic drugs were administered simultaneously to prevent gastrointestinal ulcers, hypertension and bleeding diathesis.
We agree; we added a comment regarding this issue in the results section.
Please see bolded and underlined.
On behalf of all the authors,
Sincerely yours
The corresponding author

Reviewer 2 Report (New Reviewer)
The paper is interesting, but it comes with a hasty draft.
Since it can be of considerable interest to those who deal with ALRP even in non-specialist centers, it is important to clarify some passages better.
-page 5 In the Method, you talk about CT and MRI as recent exams, without specifying anything else. It would be useful to explain (if so) that it was the tests that led to the formulation of Table 1, which lists the status of relationships of the lumbar spine and root structures.
-page 6 The administration and dosages of Dexamethasone are always the same and continuous up to a maximum of 14 days. Have you ever had an administration with progressive decrease?
As for the dosages, it is certainly known that dexamethasone, in emergency conditions (example: cerebral edema) can be administered in very high dosages.
However, since it is a powerful glucocorticoid, even a minimal but continuous dosage of 0.5 mg / kg per day can lead to stress on the adrenal glands.
It would be important to report if antacid and anticholinergic drugs were administered simultaneously to prevent gastrointestinal ulcers, hypertension and bleeding diathesis.
Author Response
August 27, 2022
REVISION COVER LETTER
I am enclosing the revised manuscript entitled "Intravenous corticosteroid therapy for acute lumbar radicular pain" submitted to the "Journal of Clinical Medicine" after extensive revision based on your reviewers' remarks for possible evaluation.
With this manuscript's submission, I would like to undertake that the manuscript mentioned above has not been published elsewhere, accepted for publication elsewhere or under editorial review for publication elsewhere. My Institute's representative is fully aware of this submission.
Type of Submitted manuscript:
- Original Article
I want to share the following information with Editor-in-Chief:
"The efficacy of pharmacological interventions for acute lumbar radicular pain (ALRP) is limited, and systemic steroid use remains controversial. We evaluated the effectiveness and tolerance of systemic steroid use in a cohort of patients with ALRP.
In our retrospective cohort of 56 patients (24 females) admitted with intractable ALRP resistance to conservative treatment, all patients received a daily dose of IV 24 mg Dexamethasone until discharge, SNRB or Surgery.
The average IV Steroid treatment was 3.9 (±2.8) days, with most showing significant improvement enough to merit discharge (69.7%). SNRB was required in 19.6% and surgical intervention in 10.7% within the same admission. Multivariate analysis did not find any parameter to predict treatment failureand one patient required discontinuation of IV steroids due to elevated blood pressure.
Despite the insufficient evidence in the literature, IV steroid treatment is still a viable option in ALRP treatment with good pain relief (70%) and a low complication rate. Even today, medical resources are limited worldwide, and portions of the population do not enjoy accessibility to specialized pain clinics. A simple and available treatment protocol that provides an early treatment option for ALRP with similar results to SNRBs is still required. The once-daily treatment protocol of intravenous 24mg Dexamethasone is simple (one dose for all), available, cheap, and without significant adverse events achieving 70% of early pain relief. This protocol is ideal for general practitioners, rural areas, and hospitals lacking pain services and supports the common practice used in many spine units."
We believe this work's significant findings can contribute to the ongoing debate on the subject and add further knowledge and experience to the world community. As such, we would like them to be published in the journal.
Attached to this letter is our point-to-point reply below to the reviewers' comments with changes marked in our revised manuscript submitted.
On behalf of all the authors,
Sincerely yours
The corresponding author
Dear Editor
Please see our point-to-point reply below to the reviewers' comments with changes marked in our revised manuscript submitted.
We would like to appreciate the reviewer's remarks and efforts in helping us make this work better.
Review reports:
Reviewer 1 & Replies to review:
- Inclusion criteria: The caption of the paper is ACUTE lumbar radicular pain. What is the definition of "ACUTE"? Does it include patients who attend the ER within hours after the onset of pain? There is a mention that CT/MRI must have been done within 6 months, which makes the definition of acute even more confusing. If the patient has pain for several months before going to the ER (or may have gone to ER more than once), calling it acute lumbar radiculopathy may not be appropriate. In a similar paper by Kovarsky (reference 27) this controversy has been avoided by using the term lumbosacral radiculopathy without qualifying it as acute. The authors could provide the duration of pain (from the onset to the date of entry into the study) in the table and see if t contributed to the outcome after steroid therapy.
We included all patients with acute ongoing lumbar radicular pain lasting from the same day to six months, as described in the methods.
We have made this point clearer now in the text.
Change is bolded and underlined.
Failure of treatment in the ER was considered as severe acute lumbar radicular pain. Unfortunately, accurate data regarding pre-hospital symptom duration is lacking. Therefore analysis of its impact on steroid treatment results is unavailable.
- The choice of dose of Dexamethasone appears to be purely empirical. How did the surgeons determine that 20 mg IV is the ideal dose without trying other doses and establishing it as the protocol for your clinic?
We agree 24 mg IV dose of Dexamethasone is an empirical historical dosage used in our department for many years. We studied the results of this protocol.
Reviewer 2 & Replies to review:
The paper is interesting, but it comes with a hasty draft.
Since it can be of considerable interest to those who deal with ALRP even in non-specialist centers, it is important to clarify some passages better.
-page 5 In the Method, you talk about CT and MRI as recent exams, without specifying anything else. It would be useful to explain (if so) that it was the tests that led to the formulation of Table 1, which lists the status of relationships of the lumbar spine and root structures.
Please see the methods paragraph as attached here:
Two fellowship-trained spine surgeons reviewed the cohort's imaging studies (CT and/or MRI available) and classified findings by etiology (Herniated disc versus degenerative stenosis), maximal area of nerve compression (central canal, recess, foraminal and extraforaminal) and the degree of pressure (contact, light pressure, severe pressure). All parameters were scored by three consultants (two experienced spinal surgeons and one musculoskeletal radiologist).
Also:
In the results section, we added a referral to the results gathered from those studies in Table 1.
-page 6 The administration and dosages of Dexamethasone are always the same and continuous up to a maximum of 14 days. Have you ever had an administration with progressive decrease?
The protocol is 24mg single dose daily for 2-7 days without a dose change. No patient had a progressive decrease in dosage.
As for the dosages, it is certainly known that Dexamethasone, in emergency conditions (example: cerebral edema) can be administered in very high dosages.
However, since it is a powerful glucocorticoid, even a minimal but continuous dosage of 0.5 mg / kg per day can lead to stress on the adrenal glands.
It would be important to report if antacid and anticholinergic drugs were administered simultaneously to prevent gastrointestinal ulcers, hypertension and bleeding diathesis.
We agree; we added a comment regarding this issue in the results section.
Please see bolded and underlined.
On behalf of all the authors,
Sincerely yours
The corresponding author

This manuscript is a resubmission of an earlier submission. The following is a list of the peer review reports and author responses from that submission.
Round 1
Reviewer 1 Report
This study is a very interesting topic on the efficacy of pharmacological interventions for acute lumbar radicular pain
(ALRP). It is a study with questionable sample size, so information on how the sample size was calculated should be provided.
Second, what were the criteria for inclusion and exclusion. Whether or not there has been blinding in the measurement process.
It should also be explained what type of study it is, as there is no control or placebo group.
The results of the study are not sufficient to be able to affirm the conclusions of the summary, it would also be necessary to provide conclusions to the study, taking into account the adverse effects.
Author Response
REVISION COVER LETTER
I am enclosing the revised manuscript entitled "Intravenous corticosteroid therapy for acute lumbar radicular pain" submitted to the "Journal of Clinical Medicine" after extensive revision based on your reviewers' remarks for possible evaluation.
With this manuscript's submission, I would like to undertake that the manuscript mentioned above has not been published elsewhere, accepted for publication elsewhere or under editorial review for publication elsewhere. My Institute's representative is fully aware of this submission.
Type of Submitted manuscript:
- Original Article
I want to share the following information with Editor-in-Chief:
"The efficacy of pharmacological interventions for acute lumbar radicular pain (ALRP) is limited, and systemic steroid use remains controversial. We evaluated the effectiveness and tolerance of systemic steroid use in a cohort of patients with ALRP.
In our retrospective cohort of 56 patients (24 females) admitted with intractable ALRP resistance to conservative treatment, all patients received a daily dose of IV 24 mg Dexamethasone until discharge, SNRB or Surgery.
The average IV Steroid treatment was 3.9 (±2.8) days, with most showing significant improvement enough to merit discharge (69.7%). SNRB was required in 19.6% and surgical intervention in 10.7% within the same admission. Multivariate analysis did not find any parameter to predict treatment failureand one patient required discontinuation of IV steroids due to elevated blood pressure.
Despite the insufficient evidence in the literature, IV steroid treatment is still a viable option in ALRP treatment with good pain relief (70%) and a low complication rate. Even today, medical resources are limited worldwide, and portions of the population do not enjoy accessibility to specialized pain clinics. A simple and available treatment protocol that provides an early treatment option for ALRP with similar results to SNRBs is still required. The once-daily treatment protocol of intravenous 24mg Dexamethasone is simple (one dose for all), available, cheap, and without significant adverse events achieving 70% of early pain relief. This protocol is ideal for general practitioners, rural areas, and hospitals lacking pain services and supports the common practice used in many spine units."
We believe this work's significant findings can contribute to the ongoing debate on the subject and add further knowledge and experience to the world community. As such, we would like them to be published in the journal.
Attached to this letter is our point-to-point reply below to the reviewers' comments with changes marked in our revised manuscript submitted.
On behalf of all the authors,
Sincerely yours
The corresponding author
Dear Editor
Please see our point-to-point reply below to the reviewers' comments with changes marked in our revised manuscript submitted.
Reviewer 1 – Replies to review:
This study is a very interesting topic on the efficacy of pharmacological interventions for acute lumbar radicular pain
(ALRP). It is a study with questionable sample size, so information on how the sample size was calculated should be provided.
Second, what were the criteria for inclusion and exclusion. Whether or not there has been blinding in the measurement process.
It should also be explained what type of study it is, as there is no control or placebo group.
The results of the study are not sufficient to be able to affirm the conclusions of the summary, it would also be necessary to provide conclusions to the study, taking into account the adverse effects.
We would like to appreciate the reviewer's remarks and efforts in helping us make this work better.
The study included all patients with ALRP admitted between 2016-2018 and treated in our department.
There was no sample size calculation as all patients were included and assessed.
From the methods section –
"Inclusion criteria included patients suffering from radicular pain in the ER, failed treatment with an intramuscular Meperidine, an intramuscular Diclofenac and an intravenous Assival with a recent CT or MRI (less than six months from the event). Those patients with non-improving ALRP were admitted for steroid treatment in our department."
Excluded patients from the study –
"Patients not suitable for steroidal treatment were excluded due to uncontrolled diabetes, hypertension, anticoagulant therapy, systemic infection, malignancy, and patients that refused steroidal treatment. Elective admissions for discectomies or selective nerve root injections were excluded as well. Patients diagnosed with cauda equina syndrome or new motor deficits (less than 48 hours) were operated on urgently, thus excluded."
As a retrospective cohort study, there was no randomization or blinding of the treatment.
Type of study from the text – See methods section -
A retrospective cohort study of all ALRP patients admitted to the Orthopedic Surgery department between the years 2016 to 2018.
We added comments within the discussion section (underlined) regarding the complications rate in this cohort and precautions due to its small cohort size.
Reviewer 2 Report
This is a small retrospective study by Hershkovich and colleagues examining an analgesic protocol by using IV dexamethasone for acute lumbar radicular pain. The evidence for the use of IV dexamethasone for acute radicular pain is low and the use of steroids can have significant side effects. Despite this, the author states this is used by many spinal units which is surprising. Although it is punchy and easy to read, the content lacks validity and it is impossible to perceive success from the IV steroid alone. Patients would have likely been on other analgesics, receiving physio and possibly would have improved with time alone. Due to the retrospective nature of the study there are no reliable indicators reported other than discharge and no follow up data after discharge. Many of the statements lack justification and referencing which I have mentioned below. There are also parts that don’t tie up thus inviting scrutiny and lack of reliability of the data. For instance, the abstract states patients were given IV steroid until discharge (what was the maximum time?) Methods then state treatment is given for a maximum of seven days. Results then state dexamethasone was given for a range of 2-14 days ???
Have added other pieces of info below
Abstract: You say all patients were given 24mg of dexamethasone until discharge did you go on forever if not discharged? this is an incredible dose of steroid.
What would merit intervention and when, if you have SNRB’s available why not do this instead, evidence base is better and is cheaper?
What do you mean by 69.7% to merit discharge – you need to give numbers here.
What do you mean by good pain relief (70%)? What is good pain relief ?
Introduction:
· You need to mention Wilby study in Lancet rheumatology 2021 comparing surgery and TF epidurals.
· You have to also note that a lot of the side effects from steroids occur later and half of the patients on prospective studies report side effects at 3 weeks. Some studies show up to 30% of patients.
· You cannot really compare hospital admission with invasive procedure done as an outpatient, if so please give evidence and reference.
· What evidence do you have that steroids increase the concentration around the HNP?
· Are IV steroids easy and accessible? This would mean hospital admission would it not? Due to the covid pandemic is this good use of resources?
· You are going to need to justify the need for IV steroids versus oral. Can you explain the rationale? Is there any data on this?
Methods- retrospective
· Please clarify why you chose and what you mean by CT or MRI was 6 months from the event ? Is CT sensitive and specific enough?- did you check that clinical features matched findings on MRI scans?
· How long were Steroid related side effects collected for?
· Why were patients admitted and not given a SNRB as an outpatient- this is surely cheaper?
Results:
· You say VAS was recorded, where is it?
· You say in methods that patients continue treatment for maximum seven days, in results you have a range of 2-14 days? Please explain
· What is significant pain relief ? 50% reduction ? NRS >2
· You state the rationale is to get steroid to the HNP but 24.3% of your patients have severe spinal stenosis- this does not make sense as pain associated with the pathology of spinal stenosis is not entirely clear and differs from radicular pain
· What about re-admissions? Did the patients need SNRB or elective surgery within the year?
· How do you know about steroid related side effects when you stopped collecting data on discharge?
· How can you tell if the steroid was the true effect of pain relief or if it was related to time or other drugs?
Discussion:
· You need to start with your key findings , what does your study say?
· Can you really state IV steroids have the same outcome as SNRB?? I do not feel you can make this statement
· RCTs have been done already and the evidence is low
Conclusion:
· Better to state the evidence surrounding the use of IV steroids still remains relatively unexplored rather than state it is used in many units. You need to justify this. Your study also does not support it as common practice and your statements are too strong.
Author Response
July 19, 2022
REVISION COVER LETTER
I am enclosing the revised manuscript entitled "Intravenous corticosteroid therapy for acute lumbar radicular pain" submitted to the "Journal of Clinical Medicine" after extensive revision based on your reviewers' remarks for possible evaluation.
With this manuscript's submission, I would like to undertake that the manuscript mentioned above has not been published elsewhere, accepted for publication elsewhere or under editorial review for publication elsewhere. My Institute's representative is fully aware of this submission.
Type of Submitted manuscript:
- Original Article
I want to share the following information with Editor-in-Chief:
"The efficacy of pharmacological interventions for acute lumbar radicular pain (ALRP) is limited, and systemic steroid use remains controversial. We evaluated the effectiveness and tolerance of systemic steroid use in a cohort of patients with ALRP.
In our retrospective cohort of 56 patients (24 females) admitted with intractable ALRP resistance to conservative treatment, all patients received a daily dose of IV 24 mg Dexamethasone until discharge, SNRB or Surgery.
The average IV Steroid treatment was 3.9 (±2.8) days, with most showing significant improvement enough to merit discharge (69.7%). SNRB was required in 19.6% and surgical intervention in 10.7% within the same admission. Multivariate analysis did not find any parameter to predict treatment failureand one patient required discontinuation of IV steroids due to elevated blood pressure.
Despite the insufficient evidence in the literature, IV steroid treatment is still a viable option in ALRP treatment with good pain relief (70%) and a low complication rate. Even today, medical resources are limited worldwide, and portions of the population do not enjoy accessibility to specialized pain clinics. A simple and available treatment protocol that provides an early treatment option for ALRP with similar results to SNRBs is still required. The once-daily treatment protocol of intravenous 24mg Dexamethasone is simple (one dose for all), available, cheap, and without significant adverse events achieving 70% of early pain relief. This protocol is ideal for general practitioners, rural areas, and hospitals lacking pain services and supports the common practice used in many spine units."
We believe this work's significant findings can contribute to the ongoing debate on the subject and add further knowledge and experience to the world community. As such, we would like them to be published in the journal.
Attached to this letter is our point-to-point reply below to the reviewers' comments with changes marked in our revised manuscript submitted.
On behalf of all the authors,
Sincerely yours
The corresponding author
Reviewer 2 – Replies to review:
This is a small retrospective study by Hershkovich and colleagues examining an analgesic protocol by using IV dexamethasone for acute lumbar radicular pain. The evidence for the use of IV dexamethasone for acute radicular pain is low and the use of steroids can have significant side effects. Despite this, the author states this is used by many spinal units which is surprising. Although it is punchy and easy to read, the content lacks validity and it is impossible to perceive success from the IV steroid alone. Patients would have likely been on other analgesics, receiving physio and possibly would have improved with time alone. Due to the retrospective nature of the study there are no reliable indicators reported other than discharge and no follow up data after discharge. Many of the statements lack justification and referencing which I have mentioned below. There are also parts that don't tie up thus inviting scrutiny and lack of reliability of the data. For instance, the abstract states patients were given IV steroid until discharge (what was the maximum time?) Methods then state treatment is given for a maximum of seven days. Results then state dexamethasone was given for a range of 2-14 days ???
We would like to appreciate the reviewer's remarks and efforts in helping us make this work better.
References about the use of IV dexamethasone for acute radicular pain were added to the introduction section. As mentioned in the manuscript, despite the lack of solid evidence, IV steroids are still commonly used to treat ALRP. We rephrased the abstract conclusion.
We have addressed the concerns regarding concomitant treatment in the manuscript discussion - paragraphed underlined in the text.
"This study included ALRP patients resistant to major initial treatment in the ER (Opiate, NSAIDS and muscle relaxant). ALRP usually resolves within weeks to months [38, 39]; studies showed ALRP improvement under rest, physiotherapy and pain control over several weeks of treatment. Adding IV steroids or SNRB can shorten the severe acute pain period [12, 13, 15]. As hospitalization beds are a valuable health resource, the treatment goal is to gain fast pain control to enable discharge. That is why an immediate IV steroid protocol was initiated to treat ALRP patients in our unit."
Endpoints to the study included – discharge as successful treatment, surgery or SNRB within the same admission as treatment failure. All patients were followed up in the outpatient clinic. We did not have re-admissions within three months of patients successfully treated with the IV steroids protocol. We emphasized this in the manuscript text (underlined results section).
Regarding concerns about the dexamethasone treatment period – we added an explanation to the time frame mentioned – "Iv Dexamethasone was continued until the patient received SNRB or surgery and up to 14 days".
Have added other pieces of info below
Abstract: You say all patients were given 24mg of dexamethasone until discharge did you go on forever if not discharged? this is an incredible dose of steroid.
As mentioned in our manuscript, IV steroids treatment was discontinued upon discharge (when improved), SNRB or surgery (when not improved). No patient was treated for more than 14 days.
What would merit intervention and when, if you have SNRB's available why not do this instead, evidence base is better and is cheaper?
SNRBs are not available on a daily based and are not suitable for patients under anticoagulation Tx.
We emphasised this in the text – "Even today, medical resources are limited worldwide, and portions of the population do not enjoy accessibility to specialized pain clinics. Other patients cannot undergo spinal injections due to anticoagulation treatment but require urgent pain relief. A simple and available treatment protocol that provides an early treatment option for ALRP with similar results to SNRBs [35, 36] is required."
What do you mean by 69.7% to merit discharge – you need to give numbers here.
Abstract results paragraph was made more explicit - "The average IV Steroid treatment was 3.9 (±2.8) days, with most patients showing significant pain relief to allow discharge (69.7%)."
What do you mean by good pain relief (70%)? What is good pain relief ?
Abstract conclusion paragraph was made more explicit - " Despite the insufficient evidence in the literature, IV steroid treatment is still a viable option in ALRP treatment, with pain relief allowing discharge in 70% of patients and a low complication rate."
Introduction:
- You need to mention Wilby study in Lancet rheumatology 2021 comparing surgery and TF epidurals.
Added to the introduction as ref.
- You have to also note that a lot of the side effects from steroids occur later and half of the patients on prospective studies report side effects at 3 weeks. Some studies show up to 30% of patients.
Short-term systemic steroids, up to two weeks, do not require tapering down and are less likely to cause side effects than prolonged treatment periods of more than three weeks.
- You cannot really compare hospital admission with invasive procedure done as an outpatient, if so please give evidence and reference.
Please see methods exclusion criteria - "Elective admissions for discectomies or selective nerve root injections were excluded as well."
- What evidence do you have that steroids increase the concentration around the HNP?
We have rewritten the text to be more precise – text underlined in the introduction.
- Are IV steroids easy and accessible? This would mean hospital admission would it not? Due to the covid pandemic is this good use of resources?
We agree that Covid time requires the best treatment to shorten admission. Iv steroids in our study gave the needed results and allowed a shorter admission time.
- You are going to need to justify the need for IV steroids versus oral. Can you explain the rationale? Is there any data on this?
We agree and added a comment on our limitation section of the study - text underlined in the discussion.
Methods- retrospective
- Please clarify why you chose and what you mean by CT or MRI was 6 months from the event ? Is CT sensitive and specific enough?- did you check that clinical features matched findings on MRI scans?
Advanced imaging, including CT or MRI of less than six months from admission, was considered relevant to the specific ALRP. Although we did not find a correlation between imaging findings and treatment outcome, as mentioned in the results section -" Further investigating of our cohort's clinical and radiographic parameters, we could not identify any parameter predicting the success or failure of IV Dexamethasone treatment for ALRP (Table 2)."
- How long were Steroid related side effects collected for?
Only while in admission as mentioned in the methods period.
- Why were patients admitted and not given a SNRB as an outpatient- this is surely cheaper?
In our country, as in most of the western world, SNRB availability is limited with a long waiting time (weeks to months).
Results:
- You say VAS was recorded, where is it?
We decided to exclude the VAS scores as the data was inconsistently documented in the EMR.
- You say in methods that patients continue treatment for maximum seven days, in results you have a range of 2-14 days? Please explain
Please see the above thought explanation.
What is significant pain relief ? 50% reduction ? NRS >2
In our study, the parameter was the patient being able to be discharged.
- You state the rationale is to get steroid to the HNP but 24.3% of your patients have severe spinal stenosis- this does not make sense as pain associated with the pathology of spinal stenosis is not entirely clear and differs from radicular pain
Spinal stenosis can result in claudication or radiculopathy, or both. The inflammatory cascade of those patients is similar to HNP patients and based on nerve irritation and edema due to mechanical pressure. Thos the IV steroids will affect both types of radiculopathy.
- What about re-admissions? Did the patients need SNRB or elective surgery within the year?
We documented a three-month re-admission rate. Please see above.
- How do you know about steroid related side effects when you stopped collecting data on discharge?
- How can you tell if the steroid was the true effect of pain relief or if it was related to time or other drugs?
We have added the explanation to the queries above, and please see above.
Discussion:
- You need to start with your key findings , what does your study say?
We have rewritten the discussion accordingly – please see in the text underlined.
- Can you really state IV steroids have the same outcome as SNRB?? I do not feel you can make this statement
Our references in that paragraph success rate of SNRB was 75-80%, comparable to our 70% with the IV steroids protocol.
- RCTs have been done already and the evidence is low
We agree with the statement, but in those publications, the number of patients is still low, and the results are inconclusive. Therefore, further studies are still needed.
Conclusion:
- Better to state the evidence surrounding the use of IV steroids still remains relatively unexplored rather than state it is used in many units. You need to justify this. Your study also does not support it as common practice and your statements are too strong.
We agree with the statement and have rewritten the conclusion in a way that correlates with your points.

Round 2
Reviewer 2 Report
Questions have not been adequately answered